

# Automatic distractor generation in multiple-choice questions: a systematic literature review

Halim Wildan Awalurahman and Indra Budi

Faculty of Computer Science, Universitas Indonesia, Depok, West Java, Indonesia

## ABSTRACT

**Background:** Multiple-choice questions (MCQs) are one of the most used assessment formats. However, creating MCQs is a challenging task, particularly when formulating the distractor. Numerous studies have proposed automatic distractor generation. However, there has been no literature review to summarize and present the current state of research in this field. This study aims to perform a systematic literature review to identify trends and the state of the art of automatic distractor generation studies.

**Methodology:** We conducted a systematic literature following the Kitchenham framework. The relevant literature was retrieved from the ACM Digital Library, IEEE Xplore, Science Direct, and Scopus databases.

**Results:** A total of 60 relevant studies from 2009 to 2024 were identified and extracted to answer three research questions regarding the data sources, methods, types of questions, evaluation, languages, and domains used in the automatic distractor generation research. The results of the study indicated that automatic distractor generation has been growing with improvement and expansion in many aspects. Furthermore, trends and the state of the art in this topic were observed.

**Conclusions:** Nevertheless, we identified potential research gaps, including the need to explore further data sources, methods, languages, and domains. This study can serve as a reference for future studies proposing research within the field of automatic distractor generation.

## INTRODUCTION

Assessment is a fundamental aspect of the educational process. It is employed to measure the extent to which learning objectives and students' abilities have been achieved (*Patra & Saha, 2019*). Assessment can be conducted in various forms and for different objectives, including formative, summative, and diagnostic assessment (*Azevedo, Oliveira & Beites, 2019*). Formative assessment is employed as part of the learning process to assist students in learning and evaluating their learning progress. Summative assessment is used to assess students' achievement of learning objectives and is typically conducted at the end of the semester. Diagnostic assessment is used to identify misconceptions among students regarding a particular topic. Of the three, formative and summative assessments are the most utilised and well-known.

Corresponding author
Halim Wildan Awalurahman,
halim.wildan@ui.ac.id

Assessment has been conducted in a multitude of formats and forms. *Das et al. (2021b)* divided the questions type into fill-in-the-blank, multiple-choice, and true/false questions. Study *Kurdi et al. (2020)* divided the type of the questions in automatic question generation research into gap-fill, wh-questions, jeopardy style, analogy, recognition, list and describe, summarise and name some, pattern-based, aggregation-based, definition, choose the type, comparison, and description. The various forms and formats have been used differently depending on the objectives and domain of the assessment (*Kurdi et al., 2020*; *Das et al., 2021b*).

One of the most used formats for educational assessment is the multiple-choice question (MCQ). MCQs have been used in a variety of contexts, including classroom examinations (*Rahmah, Yusrizal & Syukri, 2020*), national-level school examinations (*Putra & Abdullah, 2019*) and college entry tests (*Cholis & Rizqi, 2018*; *Minata et al., 2022*). MCQ was introduced in the Kansas Silent Reader Test in 1916 (*Gierl et al., 2017*). MCQs typically comprise three components: context, stem, and distractor (*Gierl et al., 2017*; *Azevedo, Oliveira & Beites, 2019*). Context refers to additional information provided to help answer the question or stem. The stem is the question itself. The distractor is a list of possible answers that the participant must select to answer the question. There is usually a key or correct answer among the distractors.

Developing multiple-choice questions (MCQs) is known to be a challenging task. The process of developing MCQs requires a deeper understanding of the subject matter and question formulation (*Vinu & Kumar, 2015*; *Azevedo, Oliveira & Beites, 2019*; *Ch & Saha, 2020*) as well as significant time and effort (*Afzal & Mitkov, 2014*; *Chiang, Wang & Fan, 2022*; *Chughtai et al., 2022*). Additionally, MCQs must satisfy certain criteria and requirements, including validity, reliability, distracting power, difficulty level, and distractor quality (*Qiu, Wu & Fan, 2020*; *Ren & Zhu, 2021*; *Rodriguez-Torrealba, Garcia-Lopez & Garcia-Cabot, 2022*). Each of these criteria presents unique challenges in obtaining the highest quality.

One of the most challenging criteria for MCQs is the presence of a suitable distractor. Ideally, a distractor should be able to confuse test takers by having a close semantic similarity to the key, sharing the same concept, or being related to the key but somehow different from it (*Gao et al., 2019*; *Bitew et al., 2022*; *Kumar et al., 2023*). If these criteria are not met, test-takers can easily differentiate the distractor and the key, reducing the effort required to answer the question. This is sub-optimal as it makes the assessment less challenging and the learning objective less measurable. It also increases the likelihood that test-takers will guess without having to use their cognitive skills in the assessment.

There has been an increase in research about automatic MCQ generation over the years (*Das et al., 2021b*). The automatic generation of questions and MCQs was created in response to the difficult process of creating questions or MCQs manually. Literature studies about automatic MCQ generation have been carried out by *Ch & Saha (2020)*, *Kurdi et al. (2020)*, *Madri & Meruva (2023)*. *Ch & Saha (2020)* has summarised the method during the MCQ generation process into procedures including preprocessing, sentence selection, answer key selection, question generation, distractor generation, and post-processing. *Kurdi et al. (2020)* also discussed automatic MCQ generation, but took a more

general view and included studies on question generation. Their study highlighted that wh-questions and gap-filling were the most commonly used formats in their primary studies. *Madri & Meruva (2023)* have similar discussions with (*Ch & Saha, 2020*) which summarized the methods into working procedures. However, they also introduced newer methods such as deep learning.

The existing studies in the literature have presented a viewpoint where distractor generation is included in MCQ generation. In contrast, many studies have separated distractor generation from MCQ generation to address the challenge of distractor generation (*Jiang & Lee, 2017*; *Gierl et al., 2017*; *Patra & Saha, 2019*; *Chiang, Wang & Fan, 2022*). This has led to the unexplored area of automatic distractor generation itself. Understanding distractor generation from a different perspective is important because of the challenge of providing distractors in MCQs (*Bitew et al., 2022*). While there may be different strategies for generating questions in MCQs, there are also many different strategies that have emerged for providing good distractors in MCQs. Some studies focus only on distractor generation, such as *Jiang & Lee (2017)*, *Gierl et al. (2017)*, *Patra & Saha (2019)*, *Chiang, Wang & Fan (2022)* which remains unexplored in the context of literature review. Therefore, the unexplored systematic literature studies on distractor generation can be considered as a research gap. Furthermore, previous literature studies have also not explored how different types of questions, data sources and methods interact with each other. The relationship between types of questions, data sources and methods in terms of which is more suitable or usable for the other has not been explored.

Based on the aforementioned research gap, this study conducted a systematic review of the literature on automatic distractor generation. The Kitchenham framework was used to conduct the review. This study aims to map the landscape of automatic distractor generation research by addressing three key questions. The first question is how each type of question interacts with data sources, languages and domains. The second question is which methods are applicable to different types of questions and data sources. The third question is which evaluation metrics are applicable to research in automatic distractor generation. This study adds a different perspective by separating distractor generation from MCQ, which highlights the studies in distractor generation. This study contributes to reporting and summarising recent trends and developments in the field of distractor generation. The results of this study can be used as a reference for future research in automatic distractor generation.

## REVIEW METHOD

This research conducted a systematic literature review (SLR) utilising the SLR framework from *Kitchenham & Charters (2007)*. The review was conducted in three stages, including planning, conducting, and reporting. This study followed the structure set out by *Awalurahman et al. (2023)* and *Fariani, Junus & Santoso (2023)*. The method section discussed the planning and conducting stages, while the report section discussed the reporting stage.

## Planning

The research questions, search strategy, and inclusion and exclusion criteria were defined during the planning stage. This study conducted a systematic literature review (SLR) to answer the following research questions:

RQ1: What is the research coverage in terms of types of questions, languages, domains and data sources?

RQ2: What methods are applicable to different types of questions and data sources?

RQ3: What evaluation metrics are applicable to research in automatic distractor generation?

In terms of the search strategy, this study utilised four databases to find relevant literature, including ACM digital library, IEEE Xplore, Science Direct, and Scopus. The keyword or query used in this research was (multiple-choice AND (distractor OR item) AND generation). All databases were set to title, abstract, and keyword search settings.

## Conducting

The conducting stage included search selection, quality assessment, and data extraction and synthesis. The search was conducted by inserting the query into the databases that had been previously determined. Inclusion and exclusion criteria were then used to filter the relevant literature. This was followed by duplicate removal. Next, abstract and title selection is performed to filter the relevant literature candidates. The candidate relevant literature is then processed into full text selection along with quality assessment to obtain the relevant literature. The relevant literature is selected as the primary source for the study. Figure 1 shows the procedure and results of each stage.

A total of 11 questions were used to assess the quality of the literature. Each question was answered on a scale of 0, 0.5 or 1, where 0 means no, 0.5 means partially and 1 means yes. The scores for all questions were then aggregated. Only literature with a cumulative score of at least six was included in this review; if the score was less than six, the literature was excluded. The questions used for the quality assessment (QA) included the following QA1: Were the objectives of the study clearly explained? QA2: Were the problem statements clearly stated? QA3: Was the source of data mentioned? QA4: Was the methodology clearly explained? QA5: Were the types of questions clearly explained? QA6: Were the distractor criteria clearly explained? QA7: Was the scoring method clearly explained? QA8: Were the scoring criteria clearly explained? QA9: Was the language mentioned? QA10: Was the domain mentioned? QA11: Were the limitations of the study clearly stated?

The literature filtered in the quality assessment stage is then extracted to identify key findings. The extraction items are shown in Table 1. Each of the extracted items should be relevant to the research questions or literature review. In addition, the quality assessment is carried out to ensure that the primary studies contain sufficient information to be extracted. The review is used to identify trends and general information.

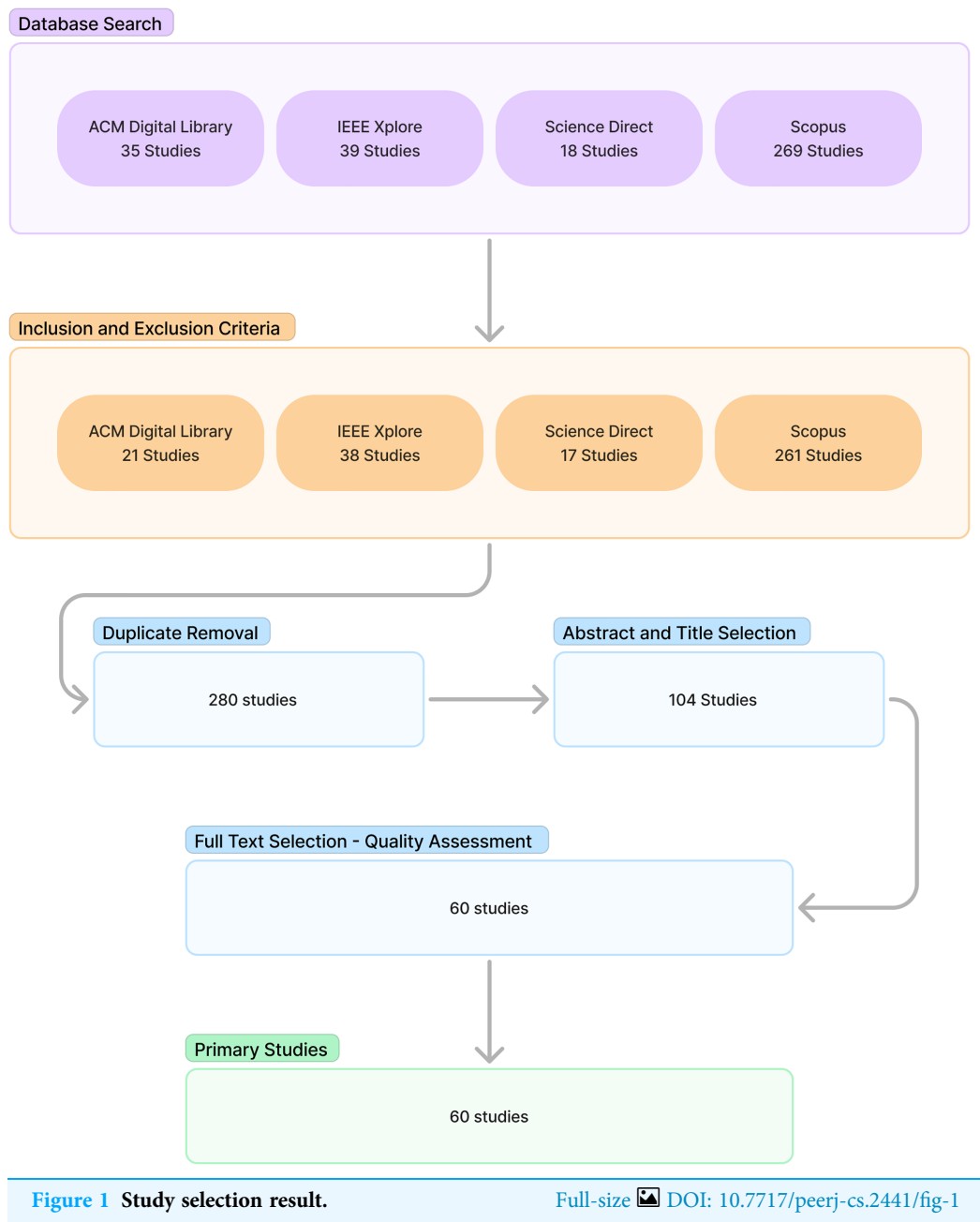

**Figure 1  Study selection result.**

## RESULTS

The results section of the study discussed the reporting phase of the SLR. This section consisted of an overview and findings for each research question. Figure 1 shows the outcome of the study selection up to the quality assessment stage. A total of 60 relevant studies were retrieved and selected as the primary source.

The result of the quality assessment is shown in Fig. 2. Each bar represents the positive value (>0) given during the quality assessment. The QA and extracted items are related. QA3, QA4, QA5, QA8, QA9 and QA10 were directly related to the data extraction form and therefore related to the research questions. Based on the results, QA1 received a

| # | Item | Description | Relevant RQ |
|---|------|-------------|-------------|
| | **Table 1** Data extraction form. | | |
| 1 | Title | | Overview |
| 2 | Authors | | Overview |
| 3 | Published date | | Overview |
| 4 | Source | Journal/Conference | Overview |
| 5 | Data source | Source of data used to generate distractor | RQ1, RQ2, RQ3 |
| 6 | Methods | Methods or technique to generate distractor | RQ2, RQ3 |
| 7 | MCQ type | Type of MCQ where distractor was generated | RQ1, RQ2 |
| 8 | Evaluation metrics | Measurement used in distractor evaluation | RQ3 |
| 9 | Language | Language used in the generated distractor | RQ1 |
| 10 | Domain | Domain used in the data source | RQ1 |

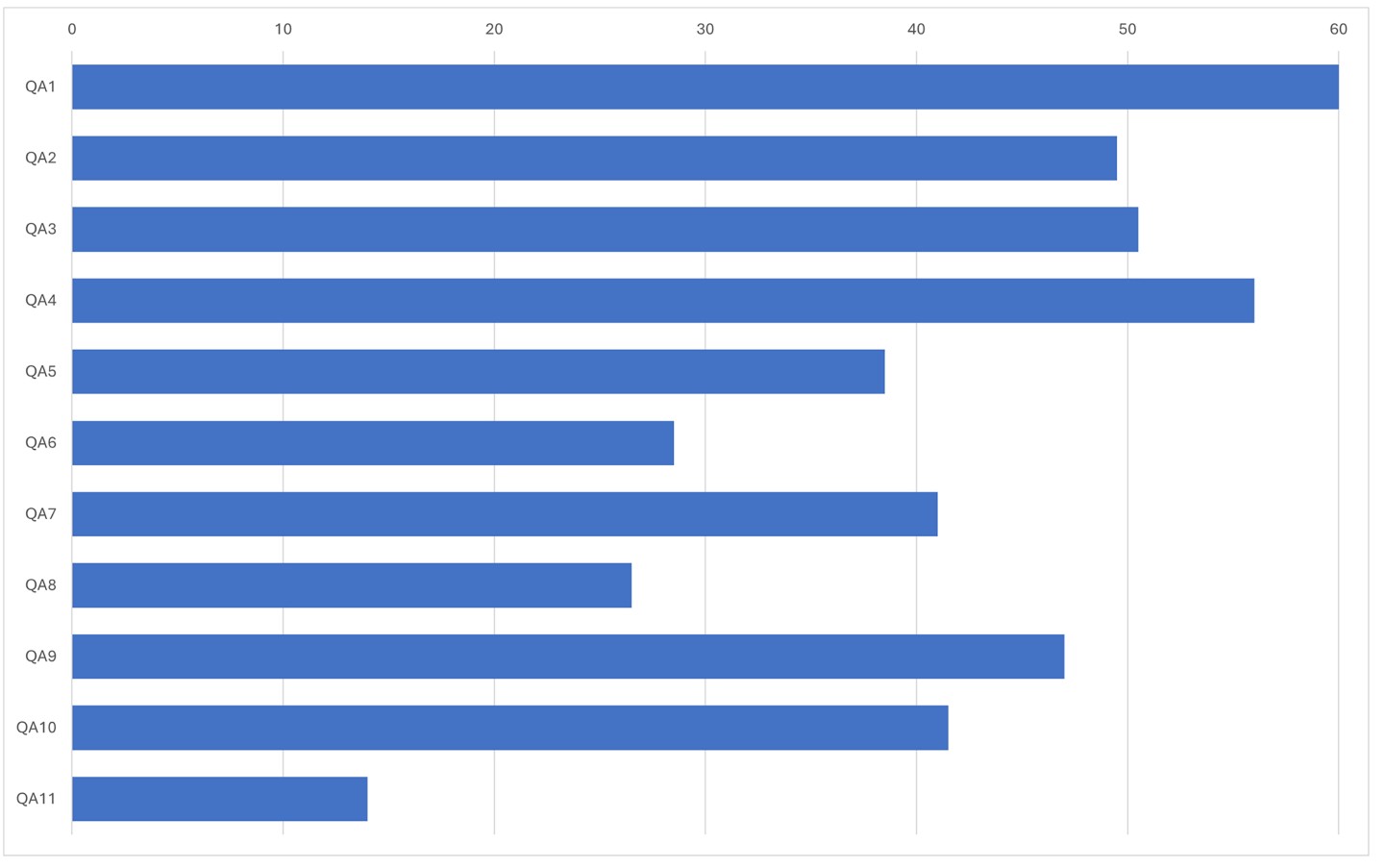

**Figure 2** Quality assessment results.

maximum score of 60. Other quality assessments, with the exception of QA5, QA6, QA8 and QA11, did not exceed 40 points. However, they are still acceptable because the quality assessment items related to the data extraction form have acceptable scores.
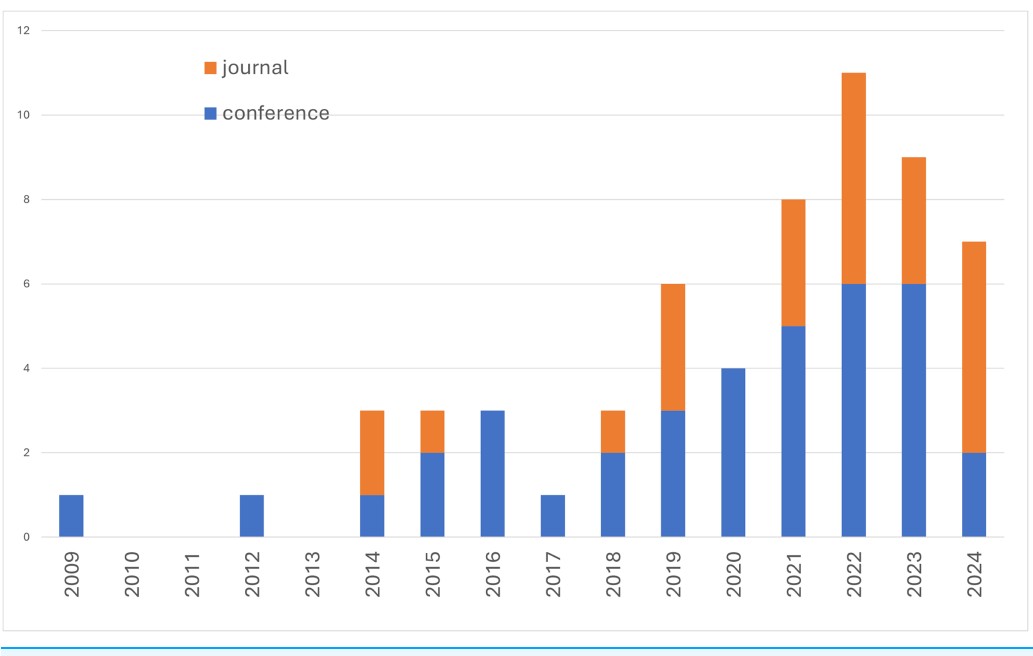

**Figure 3  Number of primary studies per year.**

## Overview

First, we reported the range of publications year within the primary studies. Figure 3 shows the number of studies per year. The earliest study we found was in 2009. The highest number of studies was found in 2022 with 11 studies. If we only consider studies published in conferences, the steady increase in publications started in 2018. Most of the studies were published in conferences, while only a few were published in journals. The most recent studies were published in 2024.

After extracting the data, in particular the types of questions, languages, domains and data sources of each primary study, we categorised each extracted item for better understanding and research mapping. Table 2 shows the categorisation with each category and description. We used guidelines from *Upadhyay et al. (2024)* to categorise the methods into three categories: linguistics, pattern matching, and statistical methods. We used (*Kurdi et al., 2020*; *Das et al., 2021b*) for the questions types categorization. While for the data source, we categorise them based on the characteristics.

## RQ1: What is the research coverage in terms of types of questions, languages, domains and data sources?

The first research question discussed the current landscape of research in automatic distractor generation. In Table 3, we use the category as the basis for the analysis, with question type and language as the rows and data source categories with domains as the columns. In terms of question type, reading comprehension is the most common format with 15 studies. Fill-in-the-blank is second with 12 studies and wh-questions is third with eight studies. Close-in-meaning was only used in three studies. The mathematical and visual formats are currently on the rise with two and three studies respectively.

**Table 2 Categorization of types of questions, data sources, and mtethods.**

| Item | Categories | Description |
|---|---|---|
| Type of questions | Closest-in-meaning | Select word that have closest meaning to the answer key |
| | Fill-in-the blank (cloze) | Select word/phrase to fill in the blank left in the stem |
| | Mathematics | Mathematics equation and calculation |
| | Mixed | Mix of others |
| | Reading Comprehension | Given context and stem to answer the questions |
| | Visual Question Answering | The questions or distractor is in form of image or visual |
| | Wh-Questions (factoid) | Does not provide context but only stem |
| Data source | *Corpus* | Unstructured text |
| | Dataset | Structured in forms of question-answer pair |
| | Knowledge base | Structured information containing entity and its relationship, *e.g.*, WordNet, Ontologies |
| Methods | Linguistic | Using language rules, grammar, semantics, *e.g.*, syntax parsing, Part of Speech. |
| | Pattern matching | Using pattern or rules syntax, *e.g.*, regular expression, knowledge base. |
| | Statistical | Using statistics for pattern or data, *e.g.*, neural networks, word embeddings. |

**Table 3 Research coverage based on type of questions, languages, domains, and data sources.**

| Count of dataset | Data source | | | | | Domain | | | | | | | | | | | |
|---|---|---|---|---|---|---|---|---|---|---|---|---|---|---|---|---|---|
| Type of questions | *Corpus* | Dataset | Know. Base | – | Grand Total | CS | Eng | Hist | Lang | Math | Med | RC | Sci | Spo | VQA | – | Grand total |
| Closest-in-meaning | 2 | | 1 | | 3 | | | | 3 | | | | | | | | 3 |
| English | 2 | | 1 | | 3 | | | | 3 | | | | | | | | 3 |
| Fill-in-the blank | 7 | 2 | 2 | 1 | 12 | 1 | 1 | 1 | 6 | | | | 2 | | 1 | | 12 |
| Bulgarian | | | | 1 | 1 | | | | 1 | | | | | | | | 1 |
| English | 1 | 2 | 1 | | 4 | | 1 | | 1 | | | | 1 | | 1 | | 4 |
| Lao | 1 | | | | 1 | | | | 1 | | | | | | | | 1 |
| Tamil | 1 | | | | 1 | | | | 1 | | | | | | | | 1 |
| Thai | | | 1 | | 1 | | | | | | | | | | 1 | | 1 |
| – | 4 | | 1 | | 5 | 1 | | 1 | 2 | | | | 1 | | | | 5 |
| Mathematics | 1 | 1 | | | 2 | | | | | | 2 | | | | | | 2 |
| – | 1 | 1 | | | 2 | | | | | | 2 | | | | | | 2 |
| Mixed | 1 | | 2 | | 3 | | 1 | 1 | | | | | | | 1 | | 3 |
| – | 1 | | 2 | | 3 | | 1 | 1 | | | | | | | 1 | | 3 |
| Reading comprehension | 2 | 12 | | 1 | 15 | | | | | | | 15 | | | | | 15 |
| English | 1 | 9 | | 1 | 11 | | | | | | | 11 | | | | | 11 |
| Portuguese | 1 | | | | 1 | | | | | | | 1 | | | | | 1 |
| Spanish | | 2 | | | 2 | | | | | | | 2 | | | | | 2 |
| Swedish | | 1 | | | 1 | | | | | | | 1 | | | | | 1 |

| Count of dataset / Type of questions | Data source | | | | | Domain | | | | | | | | | | | |
|---|---|---|---|---|---|---|---|---|---|---|---|---|---|---|---|---|---|
| | *Corpus* | Dataset | Know. Base | – | Grand Total | CS | Eng | Hist | Lang | Math | Med | RC | Sci | Spo | VQA | – | Grand total |
| Visual question answering | | 2 | 1 | | 3 | | | | | | | | | | 3 | | 3 |
| – | | 2 | 1 | | 3 | | | | | | | | | | 3 | | 3 |
| Wh-questions | 3 | 3 | 1 | 1 | 8 | 1 | | 1 | 1 | | 1 | 1 | 1 | 1 | | 1 | 8 |
| Dutch | | 1 | | | 1 | | | | 1 | | | | | | | | 1 |
| English | 1 | | 1 | | 2 | 1 | | | | | | | | 1 | | | 2 |
| – | 2 | 2 | | 1 | 5 | | | 1 | | | 1 | 1 | 1 | | | 1 | 5 |
| – | 5 | 1 | 5 | 3 | 14 | 1 | 1 | | 2 | | 3 | | 3 | | | 4 | 14 |
| English | 3 | | 2 | | 5 | | 1 | | 1 | | 1 | | 1 | | | 1 | 5 |
| – | 2 | 1 | 2 | 3 | 8 | 1 | | | 1 | | 2 | | 2 | | | 2 | 8 |
| Grand total | 21 | 21 | 12 | 6 | 60 | 3 | 3 | 3 | 12 | 2 | 4 | 16 | 6 | 1 | 3 | 7 | 60 |

**Notes:**
Domain: Computer science (CS), Engineering (Eng), History (Hist), Language Learning (Lang), Mathematics (Math), Medical (Med), Reading Comprehension (RC), Science (Sci), Sports (Spo), Visual Questions Answering (VQA), Unspecified (–).
Data Source: *Corpus*, Dataset, Knowledge Base (Know. B).

Out of 15 studies using reading comprehension format, 12 of them used dataset as the source of distractor and only two used *corpus* data. The dataset is particularly popular in this type of question since it provides formatted and structured question-answer pairs along with the context to create proper reading comprehension assessments. *Gao et al. (2019)*, *Chung, Chan & Fan (2020)*, *Shuai et al. (2023)*, *Guo, Wang & Guo (2023)* have used RACE, *Xie et al. (2022)* have used RACE along with Cosmos QA, *Shuai et al. (2023)* have used DREAM, *Dijkstra et al. (2022)* and *Qiu, Wu & Fan (2020)* have used another version of the RACE dataset. RACE dataset (*Lai et al., 2017*) is particularly popular for reading comprehension type of questions. The other data used for reading comprehension is *corpus* such as DBPedia (*Oliveira et al., 2023*).

For the fill-in-the-blank type of questions, seven studies used *corpus* as the source of the distractor. The *corpus* used consists of ESL Lounge2 (*Panda et al., 2022*), Google *Corpus* N-Gram (*Hill & Simha, 2016*), lecture slide (*Chughtai et al., 2022*), Wikipedia (*Alvarez & Baldassarri, 2018*), Vocabulary (*Murugan & Balasundaram, 2021*; *Qiu et al., 2021*) and web pages (*Das et al., 2019*). The other studies have used datasets such as CLOTH and RACE (*Wang et al., 2023*), common wrong answers (*March, Perret & Hubbard, 2021*), and knowledge base such as Probase (*Ren & Zhu, 2021*) and WordNet (*Kumar, Banchs & D'Haro, 2015*; *Kwankajornkiet, Suchato & Punyabukkana, 2016*).

The third type of question, wh-questions, used *corpus* and dataset as the data source equally often. The *corpus* used in this type of question includes ProcessBank (*Araki et al., 2016*) and Website pages (*Patra & Saha, 2019*). The dataset used in this type of question

includes NewsQuizQA (*Lelkes, Tran & Yu, 2021*), Televic and Wezooz (*Bitew et al., 2022*), and USMLE (*Baldwin et al., 2022*). Only one study used knowledge base, which is WordNet, Wiktionary, and Google Search (*Maheen et al., 2022*). The wh-questions typically do not require context. The answer could also be in the form of entities since the questions consist of who, when, and where that ask specifically about entity type.

Closest-in-meaning and mixed types were not very popular. Both used only the *corpus* and the knowledge base as a source of distractors. Study for closest in meaning type of questions used several data. Study *Yusuf, Hidayah & Adji (2023)* utilised collocation dictionaries, *Susanti, Iida & Tokunaga (2015)* used *corpus* and WordNet, while *Susanti et al. (2018)* used WordNet taxonomy along with the JACET8000 word list. For studies in mixed categories, *Kumar et al. (2023)* used WordNet and ontology and *Al-Yahya (2014)* used ontologies too.

The two remaining categories, mathematics and visual question answering were considered rising as they are recently developed. Studies in mathematics used content repositories (*Feng et al., 2024*) and reasoning datasets (*Dave et al., 2021*). The earliest studies we found for mathematics were (*Dave et al., 2021*). Studies in visual question answering mainly used datasets such as Visual7W and VQAV2 dataset (*Lu et al., 2022*; *Ding et al., 2024*) and a knowledge base with an image repository (*Singh et al., 2019*). Both studies in mathematics and visual question-answering distractor generation are still limited in terms of the number of studies.

In terms of languages, we found that nine different languages have been used in automatic distractor generation studies. The languages include English (*Gao et al., 2019*; *Chung, Chan & Fan, 2020*; *Guo, Wang & Guo, 2023*), Bulgarian (*Nikolova, 2009*), Dutch (*Bitew et al., 2022*), Lao (*Qiu et al., 2021*), Portuguese (*Oliveira et al., 2023*), Spanish (*De-Fitero-Dominguez et al., 2024*; *de-Fitero-Dominguez, Garcia-Cabot & Garcia-Lopez, 2024*), Swedish (*Kalpakchi & Boye, 2021*), Tamil (*Murugan & Balasundaram, 2021*), and Thai (*Kwankajornkiet, Suchato & Punyabukkana, 2016*). English was the most dominant language, with 25 studies, while the other languages were represented by only one or two studies each. English was most commonly used in reading comprehension type questions, together with Portuguese, Spanish and Swedish. Other languages such as Bulgarian, Lao, Tamil and Thai were used in language learning with fill-in-the-blank questions.

In terms of domains, we first had to categorise reading comprehension as a domain due to the number of studies that used the same data or generally did not include or describe the domain. We included 16 studies that used the reading comprehension domain. The second domain was language learning with 12 studies. This domain mostly consists of fill-in-the-blank questions. Most studies in this domain aim to support the language learning process by using the fill-in-the-blank type of question, which is simpler and easier to use in recognising or selecting the appropriate word or phrase given the blank in the sentence (*Murugan & Balasundaram, 2021*; *Qiu et al., 2021*; *Panda et al., 2022*). We also categorised mathematical and visual question types as their domain, as they are unique to other domains and essentially the same as the question type. For other domains, their use was spread across different question types, mostly within fill-in-the-blank and wh-questions.

In terms of data source, *corpus* and dataset were the most used with 21 studies each. Only 12 studies used knowledge base. *Corpus* data was heavily used for fill-in-the-blank questions. Fill-in-the-blank aims to eliminate a word or phrase in a sentence to leave it blank. The *corpus* was suitable for this purpose as it contains many sentences and words. In addition, the *corpus* was not formatted into a question-answer pair, which made it more flexible to choose or even randomise the position of the words to be eliminated. In contrast to the *corpus*, the data in the dataset is not so heavily used for fill-in-the-blank questions but rather dominates in reading comprehension type questions. The dataset as a source often provides the context along with the question-answer pair, which is ideal for reading comprehension tasks. Reading comprehension typically requires the reader to find relevant information in the passage before answering the question (*Day & Park, 2005*; *Lai et al., 2017*; *Dijkstra et al., 2022*), therefore the context is needed along with the question-answer pair. The knowledge base is the least used data source to generate distractors. The entity and relationship within the knowledge base are useful for fill-in-the-blank and questions where the answer is in the form of entity, synonym, hyponym, or any related words related to the answer key (*Vinu & Kumar, 2015*; *Stasaski & Hearst, 2017*; *Deepak et al., 2019*; *Maheen et al., 2022*).

Different types of questions used different types of data sources more frequently. Reading comprehension type questions most often use datasets as a data source to generate distractors. Fill-in-the-blank type questions mostly used *corpus*, similar to the closest meaning. The wh-type questions used *corpus* and dataset equally often. The mathematical and visual question types are still at an early stage of development with a small amount of research. However, current research consistently relies on datasets for both types of questions. Our findings suggest that question types interact differently with different data sources. Some types of questions require the format and characteristics of a particular data source, which makes a particular data source more suitable for certain types of questions. Each data source has its domain and language, which adds to the characteristics of the questions and distractors generated. Although there are studies that did not specify the type of questions, we have included them for future reference.

## RQ2: What methods are applicable to different types of questions and data sources?

There are three different methods to generate distractors including linguistic, pattern matching, and statistical. Table 4 shows the number of interactions between methods and data sources. The linguistic method relies on using linguistic properties such part of speech tagging (*Hill & Simha, 2016*), collocation (*Yusuf, Hidayah & Adji, 2023*), synonyms (*Huang et al., 2014*), homonyms (*Qiu et al., 2021*), hyponyms (*Huang et al., 2014*), named entities (*Oliveira et al., 2023*), and topic modelling (*Shin, Guo & Gierl, 2019*). Seven studies considered to use the linguistic methods. Pattern matching is more dependent on rule-based or patterns to extract certain information from structured data such as ontologies and WordNet. The methods in pattern matching include WordNet (*Susanti, Iida & Tokunaga, 2015*), pattern search (*Seyler, Yahya & Berberich, 2017*), synsets tree (*Kwankajornkiet, Suchato & Punyabukkana, 2016*), rule-based (*Dave et al., 2021*),

**Table 4  Number of interactions between methods and data sources.**

| Method–Type of questions | Data source | | | | |
| --- | --- | --- | --- | --- | --- |
| | *Corpus* | Dataset | Knowledge base | – | Grand total |
| Linguistic | 6 | | | 1 | 7 |
| Closest-in-meaning | 1 | | | | 1 |
| Fill-in-the blank | 2 | | | 1 | 3 |
| Reading comprehension | 1 | | | | 1 |
| – | 2 | | | | 2 |
| Pattern matching | 5 | 1 | 8 | | 14 |
| Closest-in-meaning | 1 | | | | 1 |
| Fill-in-the blank | 2 | | 2 | | 4 |
| Mathematics | | 1 | | | 1 |
| Mixed | | | 1 | | 1 |
| Visual question answering | | | 1 | | 1 |
| Wh-questions | 1 | | 1 | | 2 |
| – | 1 | | 3 | | 4 |
| Statistical | 9 | 19 | 4 | 5 | 37 |
| Closest-in-meaning | | | 1 | | 1 |
| Fill-in-the blank | 3 | 1 | 1 | | 5 |
| Mathematics | 1 | | | | 1 |
| Mixed | 1 | | 1 | | 2 |
| Reading comprehension | 1 | 12 | | 1 | 14 |
| Visual question answering | | 2 | | | 2 |
| Wh-questions | 2 | 3 | | 1 | 6 |
| – | 1 | 1 | 1 | 3 | 6 |
| – | 1 | 1 | | | 2 |
| Fill-in-the blank | | 1 | | | 1 |
| – | 1 | | | | 1 |
| Grand total | 21 | 21 | 12 | 6 | 60 |

**Note:**
 Unspecified (-)

ontology model manipulation (*Al-Yahya, 2014*), tagger (*Afzal & Mitkov, 2014*), granular ontologies (*Deepak et al., 2019*), reduce-node-label (rnl) set (*Vinu & Kumar, 2015*), graph parser (*Singh et al., 2019*), and event graph (*Araki et al., 2016*). There are 14 studies considered using pattern-matching methods. The last method is the statistical method, which relies on statistics and calculations of the data properties such as patterns, weights, and embeddings. The methods in statistical includes deep learning (*Maurya & Desarkar, 2020*; *Shuai et al., 2023*), language model (*Dijkstra et al., 2022*; *Le Berre et al., 2022*; *Wang et al., 2023*), word embedding (*Das et al., 2021a*; *Murugan & Sadhu Ramakrishnan, 2022*), sentence embedding (*Chughtai et al., 2022*; *Saddish et al., 2023*), transformers (*Lelkes, Tran*

& Yu, 2021; Foucher et al., 2022), concept embedding (Ha & Yaneva, 2018), semantic similarity (Patra & Saha, 2019), and large language model prompting (Feng et al., 2024; Grévisse, Pavlou & Schneider, 2024; Kıyak et al., 2024; Lin & Chen, 2024). There are 37 studies considered using statistical methods.

The linguistic methods mostly process *corpus* data. Six studies used *corpus* data with linguistic methods. Two of them are used in the fill-in-the-blank type of questions, utilizing part of speech tagging (Hill & Simha, 2016) and vocabulary (Qiu et al., 2021). Another study targets reading comprehension by using part of speech and named entities (Oliveira et al., 2023). The last study is used in the closest in-meaning type of questions with collocation and part-of-speech tagging (Yusuf, Hidayah & Adji, 2023). Most studies that utilized linguistic methods used part of speech tagging. Additionally, word relationships *via* vocabulary or WordNet could also be employed along with part of speech tagging. The *corpus* data is being utilized by exploring or adding the part of speech tagging. Certain parts of speech tagging patterns can be used as distractors or answer keys. Additionally, using the rule-based, the sentence could be converted into questions using the part of speech tagging too (Nikolova, 2009).

The pattern matching methods often utilized knowledge base as the source of data. Pattern matching is useful when extracting information or entities within the knowledge base (Al-Yahya, 2014; Kwankajornkiet, Suchato & Punyabukkana, 2016; Deepak et al., 2019). One particular source, WordNet, has been used in many research, especially in the fill-in-the-blank type of questions. WordNet consists of networks connecting words and entities. In fill-in-the-blank, having a distractor that is close or related to the answer key is a very important aspect (Alvarez & Baldassarri, 2018; Ren & Zhu, 2021). Finding candidate distractors this way is easier and faster. However, the knowledge base in general requires more time and effort during the development stage (Vinu & Kumar, 2015; Stasaski & Hearst, 2017). Pattern matching could also be useful in *corpus* datasets when utilizing pattern search (Seyler, Yahya & Berberich, 2017; Das et al., 2019) or event graph (Araki et al., 2016). However, this method still requires some preprocessing stages towards the *corpus* to have certain properties that can be manipulated or processed using pattern-matching tools or methods. For example, Susanti, Iida & Tokunaga (2015) utilized part of speech tagging along with co-occurring words to further use lexical hierarchy in WordNet for the distractor generation.

The statistical methods mostly process datasets as the data source. Thirty-seven studies used statistical methods, and 19 of them used the dataset as the data source. Embeddings and deep learning are most commonly found in this method category. The dataset is popular among supervised methods such as transformers (Foucher et al., 2022; Le Berre et al., 2022; Saddish et al., 2023; De-Fitero-Dominguez et al., 2024) and deep learning (Maurya & Desarkar, 2020; Shuai et al., 2021, 2023), where the input text should be formatted and the expected output is prepared. One instance is demonstrated in Rodriguez-Torrealba, Garcia-Lopez & Garcia-Cabot (2022) which used the T5 model to generate question-answer pairs along with the distractor using the dataset in a supervised fashion. The rise of the transformer model perhaps accelerates the research in automatic distractor generation too, as we saw an increase in research after the transformer

model was introduced (*Vaswani et al., 2017*). Besides transformers and pre-trained language models, the latest research has begun to utilise language model prompting (*Feng et al., 2024*; *Grévisse, Pavlou & Schneider, 2024*) to generate distractors and questions. We found five studies that have explored language prompting methods. The interaction between the instructor and the model, typically GPT (*Olney, 2023*; *Feng et al., 2024*), was able to produce questions and distractors. This approach does not require data sources as the model generates the text using its internal capabilities. Instructors typically only used descriptions or study cases to stimulate the model. The candidate distractor and questions are extracted from the model's internal knowledge.

Having analysed the interactions between data and methods, we then map the interactions between methods and data for each type of question. Our initial findings suggest that not all types of questions can be produced with certain methods. Table 4 shows that the linguistic method could only produce three types of questions, while pattern matching could produce six types of questions, and the statistical method could produce seven types of questions. This suggests that certain types of questions require certain techniques and processes that could only be achieved by a particular model.

Reading comprehension type of questions has only been produced using linguistic and statistical methods but not with pattern matching. One of the standards in reading comprehension questions is that the answer should be found within the context (*Maurya & Desarkar, 2020*; *Rodriguez-Torrealba, Garcia-Lopez & Garcia-Cabot, 2022*). Therefore, there is a minimal urgency to extract the information elsewhere. Therefore, pattern matching is not exactly the right method to generate reading comprehension questions or distractors. Even when the candidate distractor could be found using the word relationship, the candidate should have semantic similarity with the context (*Maurya & Desarkar, 2020*; *Dijkstra et al., 2022*), which might be harder to achieve using pattern matching. Additionally, pattern matching typically produced shorter, more entity-oriented distractor candidates, while reading comprehension sometimes requires longer distractor (*Maurya & Desarkar, 2020*), more of like a sentence. This gap is difficult to achieve using pattern matching but is still possible using linguistic and statistical.

Visual question answering type of questions have only been produced using pattern matching and statistical methods. No linguistic method has achieved this type of question. Visual question answering requires the model or method to understand the context of the given image, either as context or as an answer key (*Lu et al., 2022*). Linguistic method could not perform this requirement as it can only deal with text. Current visual question answering using pattern matching employs a scene graph parser (*Singh et al., 2019*) to get the relevant image from the repository while statistical methods employ multi-layer perceptron (*Lu et al., 2022*) and VL-T5 (*Ding et al., 2024*) to generate semantically correct and relevant distractor as text. Both (*Lu et al., 2022*; *Ding et al., 2024*) have to make sure that the model understands the context of the given image as the context before generating the distractor.

Mathematics questions have also been produced using pattern matching and statistical methods. Most of the mathematics questions and distractors are in the form of numbers and equations, and not regular text. Study *Dave et al. (2021)* utilized pattern matching in

the form of rule-based to generate the distractors. The distractors were generated by purposely changing the steps or formula when solving an equation. In statistics methods, study *Feng et al. (2024)* prompted the GPT 4 model to generate the mathematical questions and distractors. So, it is unlike the supervised generation with a pre-trained language model that utilized a certain dataset and target output. Instead, they used prompt engineering to generate the questions and distractors.

Wh-questions were rather surprising as they can only be achieved by statistical and pattern-matching methods. Linguistic methods, despite having been used to produce reading comprehension distractors, have not been utilized for this type of question. Wh-questions typically require shorter answers or entities as the answer (*Bitew et al., 2022*; *Maheen et al., 2022*). Linguistic methods excel in utilizing text properties such as part of speech tagging. However, to find a relevant distractor, more process is needed, including understanding the questions given and finding the most appropriate distractor that is close and semantically correct with the question (*Susanti et al., 2018*; *Lelkes, Tran & Yu, 2021*). Therefore, statistical and pattern matching has the advantage of finding the relationship between the question and candidate distractor *via* entity relations and statistical calculations such as word embedding and semantic similarity.

Different question types interact with different methods and data sources. The statistical method is the most commonly used in automatic distractor generation research, especially in reading comprehension, wh-questions and fill-in-the-blank question types. With recent advances in deep learning and transformer models, many researchers have used them for automatic distractor generation. The latest large language prompts have also been introduced into the research domain. Despite being outnumbered by statistical methods, linguistic and pattern matching methods have their advantages. Most linguistic and pattern-matching methods are faster and require less training than statistical methods. However, some methods, such as ontology model manipulation and pattern search, may depend on the quality of the available data, in this case, the knowledge base. Regardless, each method interacts differently with different types of questions and data sources.

## RQ3: What evaluation metrics are applicable for research in automatic distractor generation

In the field of distractor generation research, the generated distractors are evaluated to measure their quality and usefulness. We extracted different metrics used to evaluate the distractor. We found the top 10 most commonly used metrics in distractor generation studies. The top 10 common evaluation metrics include accuracy, difficulty, fluency, readability, relevance, precision, recall, BLEU, ROUGE and METEOR. The top 10 common evaluation metrics can be divided into two categories: those that require human annotation and those that are fully automatic. Metrics that require human annotation include accuracy, difficulty, fluency, readability and relevance. Precision and recall can fall somewhere between requiring human annotation and fully automatic, as they rely on the test data that is often generated by human annotation. For the time being, we will include them as requiring human annotation. The fully automated metrics include BLEU, ROUGE and METEOR. Fully automatic metrics are highly dependent on the training and test data.

The metrics have different uses and objectives during evaluation. Accuracy measures the percentage of the correctly generated distractor compared to the total number of distractors (*Deepak et al., 2019*; *Das et al., 2021a*; *Lu et al., 2022*). Difficulty measures the level of difficulty of the generated distractor in terms of how hard it is for the test-takers to differentiate between the key answer and the distractor (*Susanti, Iida & Tokunaga, 2015*; *Maheen et al., 2022*). Fluency decides the natural and coherent level of the generated distractor based on the language semantics and syntactic structure (*Shuai et al., 2021*; *Qiu et al., 2021*). Readability evaluates the language quality and grammar of the generated distractor (*Patra & Saha, 2019*). Relevance evaluates how relevant is it between the context, key, and distractor. Good distractor should have high relevancy with the context and key answer (*Afzal & Mitkov, 2014*). Precision measures the precision of the generated distractor by comparing the number of relevant distractors generated to the total number of distractors generated (*Bitew et al., 2022*; *Chiang, Wang & Fan, 2022*). Recall measures the accuracy of the distractor generated by comparing the number of relevant generated distractors to the total number of relevant distractors (*Deepak et al., 2019*; *Bitew et al., 2022*). Precision and recall could be important when the number of distractors is decided, and the test set contains the desired distractors.

BLEU, ROUGE, and METEOR are typically used to evaluate the statistical model or methods such as deep learning and transformers where the text generation process is involved. BLEU or Bilingual Evaluation Understudy measures the similarity between the generated distractor and the reference (*Chung, Chan & Fan, 2020*; *Rodriguez-Torrealba, Garcia-Lopez & Garcia-Cabot, 2022*). ROUGE or Recall Oriented Understudy for Gisting Evaluation measures recall from the generated distractor by comparing the n-gram overlap and word order of the generated distractor and the example (*Chung, Chan & Fan, 2020*; *Rodriguez-Torrealba, Garcia-Lopez & Garcia-Cabot, 2022*). METEOR is a machine translation evaluation metric based on the harmonic mean of precision and recall, with recall weighted more than precision (*Maurya & Desarkar, 2020*; *Dijkstra et al., 2022*; *Guo, Wang & Guo, 2023*).

The number of studies using the evaluation metrics is shown in Table 5. For a detailed analysis, we analyse the interaction for each method and dataset. It can be observed that BLEU and ROUGE are the most used metrics. Both metrics have been used in statistical methods. METEOR, although not used as often, is also only used in statistical methods. This shows that BLEU, ROUGE and METEOR are special metrics that are only used in statistical methods, while the other methods have been used in other metrics than them. Unfortunately, the extracted metrics were only from studies that used pattern matching and statistical methods. No metrics were extracted from linguistic methods. All metrics are applicable to statistical methods.

In addition to BLEU, ROUGE and METEOR, other metrics have been used several times. Fluency is the most used metric after BLEU and ROUGE with eight studies. This means that there is a lot of interest in the fluency of the generated distractor. Fluency has mostly been used in statistical methods, which means that the issue of fluency occurs in the machine-generated distractor, or distractor that involves a generation process from the model, such as the T5 transformer (*Saddish et al., 2023*) and deep learning (*Shuai et al.,*

**Table 5 Number of evaluation metrics usage in different methods and data source.**

| Count of eval metrics<br>Methods and data | Evaluation metrics<br>Accuracy | Difficulty | Fluency | Precision | Readability | Recall | Relevance | BLEU | ROUGE | METEOR |
|---|---|---|---|---|---|---|---|---|---|---|
| Pattern matching | 1 | 2 | | 3 | 2 | 3 | 1 | | | |
| *Corpus* | | | | 1 | 1 | 1 | 1 | | | |
| Knowledge base | 1 | 2 | | 2 | 1 | 2 | | | | |
| Statistical | 3 | 4 | 7 | 2 | 1 | 3 | 5 | 12 | 10 | 5 |
| *Corpus* | 1 | 1 | | 1 | 1 | 1 | 1 | 1 | | |
| Dataset | 2 | | 5 | 1 | | 2 | 3 | 11 | 10 | 5 |
| Knowledge base | | 1 | | | | | | | | |
| – | | 2 | 2 | | | | 1 | | | |
| – | | 1 | 1 | | | | | | | |
| *Corpus* | | 1 | | | | | | | | |
| – | | | 1 | | | | | | | |
| Grand total | 4 | 7 | 8 | 5 | 3 | 6 | 6 | 12 | 10 | 5 |

*2023*). Difficulty is a metric that has been used in both pattern matching and statistical methods. This means that a difficult distractor is very important in both methods. Relevance is another metric used in both methods. In addition to being difficult, distractors should also be relevant to the context and the questions.

Deciding which metrics to use is an important part of conducting automatic distractor generation studies. The top 10 metrics we have presented can be a reference for future studies. Not all 10 metrics should be used, but choosing the most appropriate one is very important. For example, when dealing with statistical methods, it might be best to use metrics such as BLEU, ROUGE, METEOR and fluency. In other cases, other metrics should be considered for evaluation.

## DISCUSSION

The discussion section is divided into smaller subsections discussing general findings, each research question, challenges and future directions, and limitations.

### General findings

The first finding of this study is the landscape of automatic distractor generation research, including the types of questions, data sources and methods. The finding is shown in Fig. 4. Sixty (60) studies were selected as primary sources in this research, ranging from 2009 to 2024. The finding summarises the types of data sources, questions and methods. Each data source has its domain and language. The data sources were divided into three categories: *corpus*, dataset and knowledge base. There are seven types of questions, including close-in-meaning, fill-in-the-blank, mathematics, mixed, reading comprehension, visual question

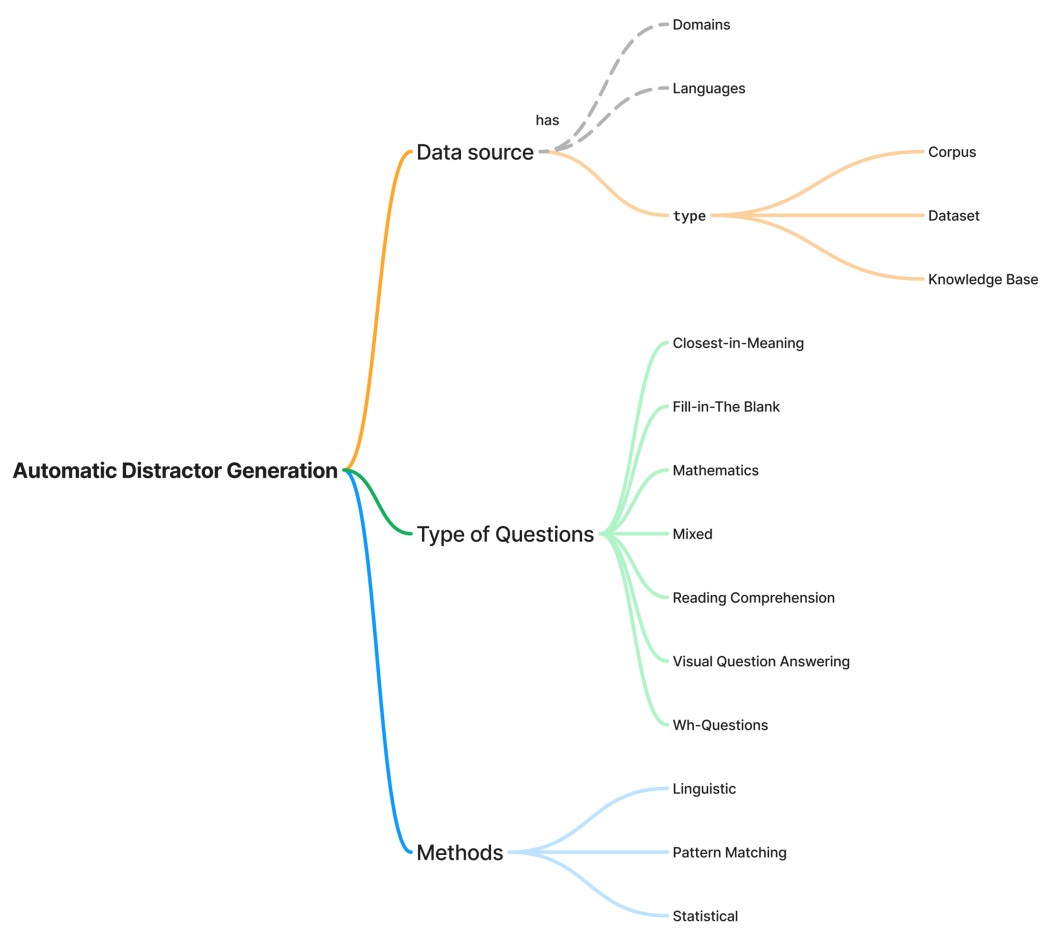

**Figure 4 Mind map of automatic distractor generation research.**

answering, and wh-questions. The methods used for automatic distractor generation can be categorised into three groups including linguistic, pattern matching and statistical.

## RQ1: What is the research coverage in terms of types of questions, languages, domains and data sources?

Different types of questions have preferred data sources that come with their languages and domains. Domains and languages are often predetermined by the data source. For example, studies using RACE as data (*Maurya & Desarkar, 2020*; *Rodriguez-Torrealba, Garcia-Lopez & Garcia-Cabot, 2022*) will have English and reading comprehension as the language and domain, since it is already given in the data. However, some studies have been able to manage these properties by using certain techniques such as translation. One example is given by *Kalpakchi & Boye (2021)* which translated the SQuAD dataset (*Rajpurkar et al., 2016*) which was originally written in English into Swedish. Another example is given by *De-Fitero-Dominguez et al. (2024)* which decided to use another version of RACE specifically built for distractor generation tasks. Different types of questions also serve different purposes. Fill-in-the-blank questions are often used for language learning because it is easier to understand the language by separating and

matching the words or tokens, as shown in this type of question (*Kwankajornkiet, Suchato & Punyabukkana, 2016*; *Murugan & Sadhu Ramakrishnan, 2022*).

Visual question answering and mathematics types of questions are considered newcomers since there is not many research on these types of questions. Although *Das et al. (2021b)* have mentioned visual question generation, the studies we found generated distractors in addition to questions and key answers. In other words, the visual multiple-choice question was only discussed in our study. Mathematics type of questions have not been explored in the previous literature studies (*Ch & Saha, 2020*; *Kurdi et al., 2020*; *Das et al., 2021b*; *Madri & Meruva, 2023*). Study *Ch & Saha (2020)* even mentioned that most automatic question generation studies avoid processing numerical characters and only focus on the text portion of the data. Therefore, our study is the first one to discover that math multiple-choice questions have been developed and could be produced.

In terms of domains, languages and question types, we believe that the current landscape is already diverse enough with many domains and question types. However, in terms of language, the number of studies is still limited. From previous literature studies (*Kurdi et al., 2020*), the notion of limited languages used has been raised. Although we have seen an increase in the number of languages compared to previous studies (*Kurdi et al., 2020*), the number of studies for non-English languages is still rare.

## RQ2: What methods are applicable to different types of questions and data sources?

Certain types of questions can be achieved using certain methods. Compared to previous studies that only categorized and listed the methods based on the category (*Ch & Saha, 2020*; *Kurdi et al., 2020*; *Das et al., 2021b*), our study mapped the number of interactions between different types of questions and methods. Our results suggest that certain methods are more useful for certain types of questions. However, we also found a category of methods that appeared to be superior to others in that they were able to produce all seven types of questions.

The statistical method is the most used approach among the primary studies. One of the most influential methods in this category is the transformer model. We found that the transformer model is capable of generating questions, answers, and distractors for fill-in-the-blank (*Wang et al., 2023*), reading comprehension (*Guo, Wang & Guo, 2023*; *De-Fitero-Dominguez et al., 2024*), visual question answering (*Ding et al., 2024*), and wh-questions (*Lelkes, Tran & Yu, 2021*; *Rodriguez-Torrealba, Garcia-Lopez & Garcia-Cabot, 2022*). Additionally, with a further improvement that turned the transformer model into an instructional large language model such as ChatGPT, it can even produce mathematics multiple choice questions *via* prompt engineering (*Feng et al., 2024*). Perhaps transformers have revolutionized the research in automatic distractor generation. This is also supported by an increasing number of research since the base transformer model was introduced in 2017 (*Vaswani et al., 2017*). Therefore, we believe that transformer models are the current state-of-the-art in automatic distractor generation method.

### RQ3: What evaluation metrics are applicable for research in automatic distractor generation

Evaluation metrics are important measures to make sure that the generated distractor is useful for the questions. Many research has discussed that currently there is no gold standard for evaluating multiple-choice questions and distractors (*Ch & Saha, 2020*; *Kurdi et al., 2020*; *Das et al., 2021b*). In the primary studies, we also found that studies used different metrics to evaluate the generated questions and distractors. To reduce the problem of evaluation metrics, we summarised the evaluation metrics across the primary studies and presented the top 10 most used evaluation metrics to be considered for evaluation in future studies. The metrics include accuracy, difficulty, fluency, precision, readability, recall, relevance, BLEU, ROUGE and METEOR. The proposed metrics can be divided into two categories: those that require a human annotator and those that are fully automated. BLEU, ROUGE and METEOR are the fully automated metrics. This division indicates that human annotation is still needed to evaluate distractor and multiple-choice questions as a whole and that we should not rely solely on automated scoring metrics (*Rodriguez-Torrealba, Garcia-Lopez & Garcia-Cabot, 2022*; *De-Fitero-Dominguez et al., 2024*).

In addition to the metrics, we also mapped the number of times studies used the metrics based on the methods and dataset. We found that all metrics are applicable in statistical methods but not in other methods, especially for BLEU, ROUGE and METEOR. However, this result is not convincing because we were not able to retrieve the metrics from studies that used the linguistic method. During the quality assessment, many studies that used linguistic methods did not specify how the distractor was scored. In addition, our decision to select only the top 10 metrics may not be representative of all the other metrics used in the primary studies.

### Limitations

The quality assessment in this study was designed and conducted by the authors, which may have introduced a degree of subjectivity that could have influenced the results. In future studies, it may be beneficial to consider incorporating external review during the quality assessment process.

### Challenges and future directions

An important finding from this literature review is that the type of question influences the method and data source. Different types of questions also influence the characteristics of the distractor. While we have learned general characteristics such as reading comprehension requiring longer distractors and fill-in-the-blank typically requiring only one word for its distractor, a deeper analysis of distractor criteria is needed. We have tried to include this in our research, but the quality assessment indicates that there is not enough evidence to answer the question of what the distractor criteria are for each type of question. From a literature review perspective, future studies focusing on literature review should attempt to analyse this question. Understanding the characteristics of the distractor needed for certain types of questions can help to improve the performance of the methods.

From the point of view of empirical studies, we suggest several future directions. First, future research can extend the current landscape of automatic distractor generation research by including more non-English languages. Alternatively, future research can also increase the number of studies in the underexplored areas. This includes conducting more research to explore underexplored methods such as large language prompting, or underexplored question types such as mathematics and visual question answering. In addition, future research could seek to develop standard evaluation procedures for multiple-choice questions, including distractor generation, which has long been a concern in this area of research.

## CONCLUSIONS

In this article, we extracted data from 60 studies on automatic distractor generation from 2009 to 2024. For the first research question, we found that different types of questions require different types of data sources to generate distractors. While the current research landscape is diverse enough in terms of question types, domains and languages, the number of studies in each aspect is still minimal. Future studies could take this opportunity to address this gap by extending the studies to more diverse types of questions, domains and languages. For the second research question, we found that statistical methods are superior because they can generate all types of questions. The most versatile method in the statistical category is the transformer method. We considered the transformer to be the state of the art model for automatic distractor generation. Future studies should consider developing more transformer models in automatic distractor generation research. In the third research question, we presented the top 10 most used evaluation metrics in the evaluation of distractor generation research. However, there is still a lack of research on metrics within linguistic methods. In addition, the evaluation standard for research in both MCQ and distractor generation has not been established. Future studies could take this opportunity to explore how to evaluate the results of MCQ and distractor generation. In addition to the outcome of the research questions, the distractor criteria and how they affect the methods were not explored in this research. Future studies could explore this issue in order to recommend which methods are more suitable for certain types of distractors. In conclusion, our study has explored the landscape of research in automatic distractor generation studies in terms of types of questions, data sources and methods.

### Funding

This work was supported by Program Penelitian Pendidikan Magister Doktor untuk Sarjana Unggul (PMDSU) under Project NKB-834/UN2.RST/HKP.05.00/2024. The funders had no role in study design, data collection and analysis, decision to publish, or preparation of the manuscript.

## Grant Disclosures

The following grant information was disclosed by the authors:
Program Penelitian Pendidikan Magister Doktor untuk Sarjana Unggul (PMDSU): NKB-834/UN2.RST/HKP.05.00/2024.

## Competing Interests

The authors declare that they have no competing interests.

## Author Contributions

- Halim Wildan Awalurahman conceived and designed the experiments, performed the experiments, analyzed the data, performed the computation work, prepared figures and/or tables, authored or reviewed drafts of the article, and approved the final draft.
- Indra Budi conceived and designed the experiments, analyzed the data, authored or reviewed drafts of the article, and approved the final draft.

## Data Availability

We used literatures as the primary data for literature review and all of them have been cited in the manuscript.

## Supplemental Information

Supplemental information for this article can be found online at http://dx.doi.org/10.7717/peerj-cs.2441#supplemental-information.

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
