# Peer review of "Automatic distractor generation in multiple-choice questions: a systematic literature review"

_PeerJ Computer Science, doi:10.7717/peerj-cs.2441_

## Round 0.1 · original submission · Major Revisions

Dear authors,

Thank you for submitting your Literature Review article. Reviewers have now commented on your article and suggest major revisions. We do encourage you to address the concerns and criticisms of the reviewers and resubmit your article once you have updated it accordingly. When submitting the revised version of your article, it will be better to also address the following:

1. The Introduction section should contain a well-developed and supported argument that meets the goals set out.
2. How this review paper will contribute to the scientific body of knowledge should be clearly mentioned.
3. The coverage (both temporal and domain) of the literature and how the literature was distributed across time domains should be clearly provided.
4. “39 stdies” in Figure 1 should be corrected.

Best wishes,

Reviewer 1 ·

Basic reporting

The paper presents a systematic review of the work carried out in the domain of distractor generation for MCQs. The review could be useful to the research community, however, there are certain improvements required in the writing of the review.
For example,
Line no. 36 to 44, the authors have mentioned about assessment, namely formative and summative, the immediate next paragraph talks about MCQs, it seems disconnected.

In the introduction, the various reviews undertaken in the past are mentioned (lines 73-88), and several limitations are highlighted. Apart from this review being a systematic review, what additional aspects are covered which were missing in the previous review. Although, the review mentions that it is focused only on distractor generation, difference from previous reviews needs to be expressed with more clarity and conviction.

Experimental design

The review systematization is accomplished through the implementation of the Kitchenham Framework. The initial survey framework is satisfactory; however, the review fails to clarify the most recent trends.

To be beneficial to the research community, it is necessary to formulate numerous profound research questions. The questions used in the review are:
RQ1: What data sources are used in distractor generation research?
RQ2: What methods are used in distractor generation research?
RQ3: What question types are used in distractor generation research?
RQ4: What evaluation methods are used for distractor generated in distractor generation
research?
RQ5: What languages have been employed in distractor generation research?
RQ6: What domains have been utilised in distractor generation research?

These questions do not offer sufficient insight into the field's progress. For it to be a research question, for example, one RQ could be to highlight what data sources are used for what type of distraction generation/question type to data source. Another question could be which methods have been applied on which languages, such kind of questions should be answered as well.

RQ1: What data sources are used in distractor generation research?
Data source category needs to be listed language wise as well as the review talks about distractor generation for different languages. Which of the sources are available in what language

RQ2: What methods are used in distractor generation research
With regards to this RQ, the Fig. no. 6 & 7 shows the various methods. The categorization can be improved, same in the text relevant should be corrected. For example, NLP and AI are very broad literature terms. Knowledge Representation falls into another category and multi-criteria and common-wrong answers, these terms are not methods. The classification hierarchy should be redesigned either methodology wise or dataset wise. It should not be a combination of both.

RQ3: What question types are used in distractor generation research
It would be better if there is a table summarizing the question types along with what types of questions have worked on what datasets and methods. This complete mapping would provide more insights to the researchers.

RQ4: What evaluation methods are used in distractor generation research
This research question is discussed in very few lines(344-348). Sufficient discussion should be provided. Although there is an accompanying table, Table No. 4, the discussion lacks depth.

RQ5 and RQ6
Here too, the discussion lacks depth.

Validity of the findings

The Discussions section has Figure 11, which is a fair summary of the review. This is a good illustration. Authors have summarized certain aspects of distractor generation, however, there should be cited references to support the authors’ perspective. There are no references in this section which needs to be corrected.
The review’s limitations are specified here. Nevertheless, the actual limitations are the reasons why the survey appears to be insufficient. As stated by the authors, "This study makes numerous significant contributions, despite its limitations." In their discussion section, they are unable to effectively substantiate this assertion.
The discussions section usually provides challenges and future directions. A research review is incomplete without this section. Discussions are meant to give critical aspects regarding the topic, this needs to be incorporated.

There is no constructive mention of future analysis and gaps in the conclusions section. Conclusions Sections should be revisited and improvised considering the above aspects.

Additional comments

Figure resolutions could be improved.
The language should be more research oriented
There should be more comparative tables rather than figures which just provide a bar graph of numbers, this will improve the readability and general interest in the manuscript’s contents.

·

Basic reporting

This paper meets the PeerJ's research domain and propose a detail literature review on distractor generation methods.

Experimental design

1. Figures need reformulation and not very clear for some of them.
2. Typos in Line 368.
3. The literature can include more state-of-the-art methods around distractor generation such as 'Distractor generation for multiple-choice questions with predictive prompting and large language models'.

Validity of the findings

No comment.

Additional comments

No comments.

---

## Round 0.2 · accepted · Accept

Dear authors,

Thank you for the revised literature review paper. One reviewer did not respond to the invitation for revision. Other reviewer thinks that you have performed the necessary additions and modifications. Your paper now seems sufficiently improved and acceptable for publication. In production step, please fix a few minor formatting and alignment issues specified by Reviewer 1.

Best wishes,

Reviewer 1 ·

Basic reporting

The manuscript now looks more readable. The added text now puts forth a more strong argument highlighting the purpose of the review.

Experimental design

The suggestions have been incorporated by the authors.

Validity of the findings

Suggestions incorporated.
Include a few references in the challenges and future directions section added by the authors recently to the manuscript.

Additional comments

Formatting and alignment of text should be checked and verified throughout the manuscript.
Table 5 is wrongly labeled Table 7
Most suggestions are incorporated in the maniscript.